# Molecular Mechanisms on the Selectivity Enhancement of Ascorbic Acid, Dopamine, and Uric Acid by Serine Oligomers Decoration on Graphene Oxide: A Molecular Dynamics Study

**DOI:** 10.3390/molecules26102876

**Published:** 2021-05-13

**Authors:** Threrawee Sanglaow, Pattanan Oungkanitanon, Piyapong Asanithi, Thana Sutthibutpong

**Affiliations:** 1Department of Physics, King Mongkut’s University of Technology Thonburi (KMUTT), Bangkok 10140, Thailand; threravee.sang@kmutt.ac.th (T.S.); pattanan.bowbow@kmutt.ac.th (P.O.); piyapong.asa@kmutt.ac.th (P.A.); 2Center of Excellence in Theoretical and Computational Science (TaCS-CoE), Faculty of Science, King Mongkut’s University of Technology Thonburi (KMUTT), 126 Pracha Uthit Rd., Bang Mod, Thung Khru, Bangkok 10140, Thailand

**Keywords:** molecular dynamics, graphene oxide, simultaneous detection

## Abstract

The selectivity in the simultaneous detection of ascorbic acid (AA), dopamine (DA), and uric acid (UA) has been an open problem in the biosensing field. Many surface modification methods were carried out for glassy carbon electrodes (GCE), including the use of graphene oxide and amino acids as a selective layer. In this work, molecular dynamics (MD) simulations were performed to investigate the role of serine oligomers on the selectivity of the AA, DA, and UA analytes. Our models consisted of a graphene oxide (GO) sheet under a solvent environment. Serine tetramers were added into the simulation box and were adsorbed on the GO surface. Then, the adsorption of each analyte on the mixed surface was monitored from MD trajectories. It was found that the adsorption of AA was preferred by serine oligomers due to the largest number of hydrogen-bond forming functional groups of AA, causing a 10-fold increase of hydrogen bonds by the tetraserine adsorption layer. UA was the least preferred due to its highest aromaticity. Finally, the role of hydrogen bonds on the electron transfer selectivity of biosensors was discussed with some previous studies. AA radicals received electrons from serine through hydrogen bonds that promoted oxidation reaction and caused the negative shifts and separation of the oxidation potential in experiments, as DA and UA were less affected by serine. Agreement of the in vitro and in silico results could lead to other in silico designs of selective layers to detect other types of analyte molecules.

## 1. Introduction

The discovery of graphene has opened the era of two-dimensional nanomaterials for different uses. Graphene is a two-dimensional sp2 hybridized carbon atom packed in a hexagonal lattice [1,2,3]. Due to its exceptionally high electrical conductivity at room temperature and its biocompatibility, graphene is regarded as a novel nanomaterial for bioelectronics and biosensing applications [4,5,6]. Chemical functionalization of graphene by amino, carboxyl, epoxy, or hydroxyl groups was carried out to improve the selectivity of carbon-based electrodes to different types of biomolecules. A possible application of the modified carbon-based substrates is the simultaneous detection of Ascorbic acid (AA), dopamine (DA), and uric acid (UA) by electrochemical techniques due to the oxidizability and the van der Waals contacts between the graphene-like structures on the surface of electrodes and the cyclic structures of these analyte molecules. AA, DA, and UA are biochemical compounds coexisting in body fluids. The deficiency of ascorbic acid is the cause of disease such as scurvy, spongy gums, ecchymosis, and petechiae [7]. Meanwhile, the lack of dopamine is related to brain disorders, such as Parkinson’s Disease, schizophrenia and Alzheimer’s [8,9]. In addition, the abnormal change of UA concentration in human body can be signs of diseases such as gout, kidney diseases, obesity, and heart diseases [10]. The goal of this wide-open research question is to precisely measure the levels of those compounds and avoid false-positive detection caused by the interference of redox potentials. To monitor the micromolar or nanomolar quantities of biomolecules for precise medical diagnosis, such techniques as ultraviolet spectroscopy [11,12], chromatography [13,14], and electrochemiluminescence [15,16] can be employed. However, the disadvantages of these methods for clinical use are the high cost and complicated procedures as the high-quality light sources or detectors are required and the measurement. Electrochemical techniques have then become popular for detecting very small amounts of molecules that could be either oxidizing or reducing agents. The relatively low costs and simple measurement procedures, requiring a three-electrode system and a potentiostat to scan for the redox potentials that return the signal of molecules, led to the possibility of developing the portable sensing platforms [17,18,19]. 

Many studies showed that using a cyclic voltammetry (CV) or differential pulse voltammetry (DPV) with glassy carbon electrodes (GCE) can detect AA, DA, and UA up to the micromolar limit of detection [20,21,22]. However, these electrochemical techniques failed to simultaneously detect AA, DA, and UA due to overlapping oxidation peaks, which might cause false-positive detection cases in clinical uses [23,24,25]. Many attempts were made to improve the selectivity by customizing the surface of electrodes with different techniques. Simultaneous AA/DA/UA detections were carried out under the presence of quinones [26], surfactants [27], and ionic liquids [28] adsorbed on the electrode surfaces. Later, multi-walled carbon nanotubes (mwCNTs) [29], graphene oxide (GO) [30], reduced graphene oxide (rGO) [20], and conducting polymers [31] were used due to stronger adsorption and chemical stability. The aforementioned techniques were used successfully for discriminating AA, DA, and UA through the separated oxidation peaks. Modifying electrodes with biopolymers, e.g., peptides, also became interested due to specific affinity for target materials and analyte molecules [32,33]. Amino acids with polar sidechain, e.g., serine with the highest density of uncharged polar groups, interact with polar functional groups of the analyte molecules through hydrogen bonds and facilitate redox reactions. Chitravathi et al. carried out a simultaneous detection of AA and DA using cyclic voltammetry with the carbon paste electrode modified by the poly-L-serine film [34], which demonstrated the capability of electropolymerized peptides as a low-cost selective layer for simultaneous detections. The same work proposed that electrostatic interactions between the deprotonated carboxyl (COO-) groups of poly-L-serine and AA in the ascorbate anion form resulted in the negative shift of the oxidation peak. A very recent experiment of our group showed that GCE modified with both GO and poly-L-serine could simultaneously detect AA/DA/UA with separated oxidation peaks (see Appendix A). Therefore, GO should also play important roles in selectivity, as was proposed by a theoretical study at a density functional theory (DFT) level in which the polar functional groups affected the interactions between substrates and analytes [35].

In this study, to gain more insight into the selective mechanisms of the polymerized L-serine and graphene oxide decorated on the GCE in a solvent environment, atomistic molecular dynamics (MD) simulations were performed for the model systems (Figure 1) consisting of a graphene oxide sheet, serine tetramers, and a number of analyte molecules. After that, the adsorption rates of the AA/DA/UA analytes were monitored along with the intermolecular contacts and hydrogen bonds. Finally, the role of hydrogen bonding on the oxidation potentials was extensively discussed in atomistic details, where intrinsic properties of serine amino acids and analyte molecules were taken into account. Understanding one of the molecular mechanisms underlying the electrochemical selectivity of modified electrodes should lead to more insight on further applications in biosensing.

## 2. Results

### 2.1. Extra Adsorption Layer of AA/DA/UA on GO with Tetraserines

Six atomistic MD simulations were performed: (a) GO + AA, (b) GO + DA, (c) GO + UA, (d) GO + SE + AA, (e) GO + SE + DA, and (f) GO + SE + UA (Figure 2). The first three structures consisted of a graphene oxide sheet and twelve AA, DA, or UA analyte molecules, while the latter also consisted of four tetraserine oligomers. In order to monitor the interaction between GO and analyte that occurred during the simulation, the minimum distance between the closest atom pairs from an analyte molecule and the GO surface was measured. All minimum distances were plotted as a function of time for each of the 12 AA, DA, and UA molecules on the GO surface without serine tetramer (Figure 3a–c) and with serine tetramers (Figure 3d–f). For all simulations, longer minimum distances were observed at the start as positions of the analyte molecules were randomized and free from binding. As the simulations progressed, minimum distances decreased when the analyte molecules became bound on the GO surface. Figure 3a displayed minimum distances between all AA molecules and the GO surface without tetraserine. Minimum distances of four AA molecules measured from GO were found about 0.25 ± 0.05 nm after 50 ns, which were about the sum of van der Waals radii of two carbon or oxygen atoms. The flat minimum distance profiles signified that the four analyte molecules stayed in close contact with GO. Meanwhile, minimum distances of eight AA molecules were alternating between the association and the dissociation states from the surface of GO. Figure 3b also showed the minimum distances between DA molecules and GO without tetraserine, in which nine DA molecules were strongly bound with GO at 0.25 ± 0.05 nm and three DA molecules were found alternating between the association and the dissociation states. In the case of UA (Figure 3c), two modes of stable minimum distances around 0.25 ± 0.05 nm and 0.55 ± 0.08 nm were found with low fluctuations. Nine UA molecules were found in closest contact with the GO surface during the last 50 ns, while two UA molecules were stabilized at the minimum distance around 0.55 ± 0.08 nm, signifying the pi–pi stacking on top of the first UA layer. Only one UA molecule was dissociated from the GO surface during the last 10 ns of the simulation.

For systems decorated with serine tetramers, tetraserines lying across the GO formed a primary layer over the surface. The tetraserine layer selectively interacts with some analyte molecules and thus shields them away from the GO surface and facilitates the stacking between analyte molecules. Figure 3d showed minimum distances between all AA molecules and the GO surface with tetraserines. There were five AA molecules bound in closest contacts with the GO surface at 0.25 ± 0.05nm, while only one AA molecule was alternating between bound and unbound states. Then, it was found that the other six AA molecules were shielded by tetraserines from the surface of GO, so that their equilibrium minimum distances were larger than the sum of van der Waals radii between two atoms. Moreover, stacking between either the ring part and the chain part of the shielded AA molecules caused the variety of equilibrated minimum distances around 0.25–1.25 nm within the last 10 ns. In the case of DA molecules binding with GO when tetraserines were present (Figure 3e), eight DA molecules were bound directly with the GO surface at the minimum distance around 0.25 ± 0.05 nm during the last 50 ns of the simulation. Another two DA molecules were either stacked on top of the tetraserines layer or the primary layer formed by the first eight DA molecules over the GO surface. Meanwhile, the last two DA molecules were alternating between the association and the dissociation states. The smaller number of bound DA on the GO surface implied that the presence of tetraserines slightly disrupted the binding of DA molecules. The binding of UA to the GO surface was almost unaffected by tetraserines. Figure 3f showed the minimum distances of the GO surface and UA molecules with the presence of tetraserines. There were nine UA molecules closely bound with GO during the last 50 ns, except by two short dissociation events. Similar to the case in which tetraserines were absent, two UA molecules were found with pi–pi stacking on top of the first layer, and another UA molecule was found dissociated from the GO surface. The similarity between the binding behavior of UA to the GO surface with and without tetraserines implied that van der Waals interactions between the GO surface and the UA molecules were significantly more important than the interactions between tetraserines and UA.

### 2.2. Serine Oligomers Unequally Affected the AA/DA/UA Adsorption on GO

Adsorption behavior of all AA, DA, and UA analyte molecules monitored in Figure 3 was then mapped onto the radial distribution function (RDF) of the minimum distance from the surfaces. The RDF profile and the coordination number (C_n_) as a function of distance from the surface were calculated between the analyte molecules and the GO surface for the systems of AA, DA, and UA adsorbed on GO without serine tetramers (Figure 4a). For the systems of AA, DA, and UA adsorbed on GO with serine tetramers, RDF and C_n_ of analyte molecules were calculated both from the GO surface (Figure 4b) and from serine tetramers (Figure 4c). The C_n_ at the first adsorption layer represented the cumulative average number N of the adsorbed analytes, and the binding constant k was then calculated in terms of N by the equation:(1)$$k=\frac{V}{m}\left(\frac{N}{N_{a}-N}\right)$$

When *k* represents the affinity of molecules, *V* is the volume of the simulation box, m is the total mass of molecules in the simulation box, *N_a_* is the total number of analytes, and *N* is the number of the adsorbed analyte molecules. The equation was related to association and dissociation rates and could relatively estimate the binding affinity of different analytes on different substrates in this study. From all cases in Figure 4, the convergence of surface RDF to zero and the convergence of coordination numbers to 12 (the total number of analytes) within the 2 nm distance suggested that most of the analyte molecules in the models were bound to the GO surface.

Figure 4a displayed the surface RDF profile of AA, DA, and UA from the GO. AA had the lowest number of molecules in the first RDF shell compared to the other two analytes, while the clear second RDF shell of UA molecules stacking on top of the first shell was clearly observed and was in concurrence with the minimum distance tracking. The coordination number (C_n_) was 8.63 for AA at the 0.48 nm distance, while the higher C_n_ of 9.93 was found for DA at the same distance. For UA, 9.73 molecules formed the first adsorption layer, and the remaining 2.36 UA molecules were adsorbed in the second layer. Binding constants were found at 0.62, 1.16, and 1.03 M^−1^ for AA, DA, and UA, respectively, suggesting that UA was the most energetically favorable analyte for the bare GO. The planar structure of UA could increase the surface area for binding and promote the van der Waals contacts. 

The absorption of analytes on the GO surface was affected by the presence of tetraserine, as shown in Figure 4b. Extra RDF shells were observed for AA, corresponding to the number of AA molecules shielded by tetraserines from the GO surface. Per the results, the C_n_ at the first adsorption layer was decreased to 4.63, and the binding constant was substantially dropped to 0.15 M^−1^. Similar to AA, the decreased first RDF peak of DA on the GO surface when decorated by tetraserines was observed along with the higher probability distribution at a further distance. This also demonstrated the shielding effects of tetraserines, but the effects on DA were less significant than those on AA. The C_n_ at the first adsorption layer for DA on the GO surface was decreased to 7.80, and the binding constant was decreased to 0.45 M^−1^. For UA, however, only a slight decrease of the first RDF peak was observed, suggesting that the binding of UA on the GO surface was mostly unaffected by tetraserine. Only a small decrease of the C_n_ at the first adsorption layer and the binding constant were found at 9.69 and 1.01 M^−1^, respectively. 

Coordination numbers and binding constants calculated from the first adsorption layers on GO were in concurrence with the analysis of minimum distances, in which AA molecules were the most affected and the UA molecules were the least affected by tetraserine decoration. Figure 4c described the adsorption of all three analytes on the surface created by tetraserine and the competing behavior with the GO surface. The first adsorption layer on a bare GO surface was defined by considering the first minimum of the RDF shells of UA at around 0.48 nm. However, for the relatively rough tetraserine surface, the first three peaks of RDF of AA and DA were accounted for the stern layer: the first peak corresponding to the sum of van der Waals radius between a pair of H atoms, the second peak corresponded to that between an H atom and a heavy (N, C, or O) atoms, and the third peak corresponded to that between heavy atoms. For UA, only the first two peaks were considered as the UA molecule was planar and only heavy atoms contributed in the RDF and C_n_. Comparing the RDF profiles of AA, DA, and UA on the tetraserine surfaces in Figure 4c, the highest first three RDF peaks representing the first adsorption layer belonged to AA, followed by DA and UA. The coordination numbers were found at 8.48, 5.10, and 3.84 for the first adsorption layers of AA, DA, and UA, respectively. Binding constants were also estimated for tetraserine binding (Table 1), and it was found that the constant 0.58 M^−1^ for AA binding was significantly higher than the 0.18 M^−1^ for DA and 0.11 M^−1^ for UA binding. 

The high affinity of AA binding on the tetraserine surface shown in Figure 4c corresponded to the lower affinity on the GO surface, illustrating the competitive behavior between tetraserine and GO surfaces on the analyte adsorption. AA molecules were most likely to bind with serine tetramers, while UA molecules were most likely to bind with GOs. Only slight differences (k_UA_/k_AA_ = 1.66 M^−1^) were found for systems without tetraserines when considering the binding constants. On the other hand, clear differences (k_UA_/k_AA_ = 6.73 M^−1^) in the binding constants were found for systems with tetraserines. These results suggested that tetraserine was unequally affected and was selective for the binding of AA, UA, and DA, which was due to the intrinsic properties of the analyte molecules. 

### 2.3. Selectivity Enhancement in Terms of Physico-Chemical Characteristics of Serine Oligomers and Analyte Molecules

In the previous section, our MD simulations and RDF analysis illustrated the selective adsorption of AA, DA, and UA analytes on the GO surface decorated by tetraserines. The selectivity was contributed by the binding preference of AA on tetraserines and UA on the GO surface. Due to a large number of polar functional groups, a tetraserine in Figure 5a tended to absorb the analytes by creating hydrogen bonds. Like other amino acids, a serine monomer consists of an amine group (N-H) and a carbonyl group (C=O) at the backbone, and a polar hydroxyl (O-H) functional group at the sidechain. As a tetramer in a zwitterion form, the positively charged N-terminal is a strong hydrogen bond donor, while the carboxyl (-COO-) group is a strong hydrogen bond acceptor. Some studies reported that the formation of hydrogen bonds caused the shifting of redox potentials [36,37,38]. Experimental work on the AA/DA/UA detections by GO substrates showed that polyserine decoration also caused some shifts of oxidation potentials. Therefore, the effect of polyserine decoration on the oxidation peaks might be explained in terms of hydrogen bond formations.

Partial charge distribution along with the atomic nomenclature of AA (Figure 5b), DA (Figure 5c), and UA (Figure 5d) displayed the positions of O-H and N-H groups, behaving as hydrogen donors or electron acceptors. The selectivity of tetraserine and its binding preference with AA could be explained in terms of molecular geometry and the distances between sites for hydrogen bonds. Consider the distances between the hydroxyl (-OH) oxygen atom of an amino acid *n* and a carbonyl (C=O) oxygen atom of the neighboring amino acids (Figure 5a). The distance between -OH at amino acid *n* and C=O at amino acids *n* + 1 and *n*−1 were found at 0.55 nm and 0.46 nm, respectively. Meanwhile, the distance between two carboxyl (COO-) oxygens was found at 0.22 nm. Now, consider the distances between hydrogen bond donors in an AA molecule (Figure 5b). The distances between O2–O5 and O2–O6 atom pairs of AA were found at 0.55 nm and 0.59 nm, respectively, and were matched with the distance between C=O and -OH of tetraserines for hydrogen bonding. In addition, the O5–O6 distance 0.32 nm was matched with the distance between the carboxylic oxygen pairs at the C-terminal of tetraserines. A similar pattern was found for the O1–O2 hydroxyl oxygen pairs of DA with a distance of 0.28 nm. Therefore, up to two hydrogen bonds could occur for each pair of an analyte and a tetraserine. For UA, all hydrogen bond donor groups were the N-H groups possessing lower polarity than the O-H groups. 

Hydrogen bond analysis was then performed between the analytes and the tetraserine decorated substrates by counting the number of events that meet these criteria: (1) the distance between a donor atom and an acceptor atom was less than 0.35 nm, and (2) bending angle made by a group of donor-hydrogen-acceptor was less than 30°. It was found that the number of hydrogen bonds between the analytes and GO in Figure 5e–g was significantly lower than the hydrogen bonds between the analytes and tetraserines in Figure 5h–j. The larger number of hydrogen bonds at tetraserines was due to the high density of functional groups mentioned earlier. The average number of hydrogen bonds between tetraserines and AA during the last 50 ns of the MD trajectory was 31.62 ± 2.87, which was higher than those for DA around 16.59 ± 1.71, and those for UA around 8.30 ± 1.96. The contribution of each hydrogen bond donor of each analyte and each oxygen atom in tetraserines as hydrogen bond acceptors was addressed through the radial distribution function of each donor around the group of acceptors atoms in Figure 6. Figure 6a showed that O6, O5, and O2 atoms were with the three highest first RDF peaks, suggesting their highest contribution for hydrogen bonding. For DA (Figure 6b), O2 was with the highest contribution, followed by O1. In the case of UA (Figure 6c), only small first RDF peaks were observed for all hydrogen bond donors with nitrogen atoms, in concurrence with the smallest number of hydrogen bonds. Conformational snapshots in Figure 6d for AA displayed the contribution of the carboxyl (COO-) oxygens at the C-terminal in binding with the O5 and/or O6 atoms of AA. Either single or double hydrogen bonds could occur at the carboxyl group. Figure 6e for DA also displays the contribution of the carboxyl (COO-) oxygens at the C-terminal as a common hydrogen bond acceptor. Either O1 or O2 of DA mostly contributed as the hydrogen bond donors, while the -NH_2_ group rarely contributed due to the weaker polarity. In addition to the hydrogen bonds with the carboxyl (COO-) oxygens of tetraserine as acceptors, the amine group at the tetraserine backbone and the hydroxyl group at the tetraserine sidechain acted as donor atoms for hydrogen bonding but were less dominant than the carboxyl (COO-) acceptors.

## 3. Discussion

In this work, atomistic molecular dynamics simulations were used to provide atomistic details on the selectivity of a GO electrode modified by serine oligomers for the simultaneous detections of the AA, DA, and UA molecules. Our model systems consisted of a small GO flake and several analyte molecules. Adsorption of analyte molecules of the GO flakes was monitored either in the presence or in the absence of tetraserines. It was found that the role of intrinsic molecular properties of the three analytes promoted the hydrogen bonding with the tetraserines, which became a selective layer. Molecular mechanisms of the enhanced selectivity by serine oligomers addressed by the MD simulations were in concurrence with our differential pulse voltammetry results of simultaneous detection of AA, DA, and UA. The largest negative shift found for the oxidation potential of AA corresponded to the highest number of hydrogen bonds with serine oligomers. Meanwhile, the smallest negative shift found for the oxidation potential of UA corresponded to the lowest number of hydrogen bonds with serine oligomers (see Appendix A). The highest number of strong hydrogen bond donors in AA resulted in combinations of hydrogen bonding patterns and the largest number of hydrogen bonds with tetraserine. Meanwhile, the weaker hydrogen bond donors and the higher aromaticity of UA corresponded to the lowest number of hydrogen bonds. The different amount of hydrogen bonds formed between tetraserine and different analytes was in concurrence with the analysis of analyte distribution about the different regions of the substrate and provided clarification on the selectivity of tetraserines. Here, the consequence of hydrogen bond formations on the oxidation potentials of the analytes would be discussed. Our assumption was made based on several reports about the changes in oxidation potentials caused by the charge transfer through hydrogen bonds [39,40,41]. A theoretical study by Fang et al. proposed that the electron transfer from a tyrosine residue to the P_680_ chlorophylls was facilitated by hydrogen bonding of the tyrosine with a deprotonated histidine residue. The phenol group of the tyrosine received electrons from the imidazole ring of the histidine and gave away the proton. After the first charge transfer, the phenol became a phenoxyl radical, for which the adiabatic ionization and the oxidation potential for the electron transfer to P_680_ became lower [42]. Analogously, the serine oligomers decorated on a GO surface possessed the very electronegative carboxyl groups, acting as hydrogen bond acceptors and electron donors. An AA analyte possessed the largest number of hydroxyl groups even in the ascorbate form at pH 7, serving as strong hydrogen bond donors and electron acceptors. Meanwhile, a tetraserine oligomer consisted of a number of both hydrogen bond donor and acceptor sites, especially a strong hydrogen bond acceptor at the C-terminus that could donate electrons to the analytes. According to the MD simulation results, the highest number of hydrogen bonds was found between the AA analytes and serine oligomers compared with DA and UA. Thus, it could be inferred that AA in the ascorbate anion received the highest amount of electrons from the serine oligomers and became an ascorbate radical that could give away electrons through one-electron oxidation, requiring lower oxidation potential than the typical two-electron oxidation of AA. This ability of AA to give away electrons, coupled with the higher probability of hydrogen bonding with serine oligomers, resulted in the negative shift of the oxidation peak observed in the experiments. For the case of DA, less amount of hydrogen bonds was observed in MD simulations compared with AA due to the smaller number of polar groups. Moreover, the oxidation of a DA in the neutral form at pH 7 was two-electron oxidation that required larger oxidation potential. As a result, the negative shifting of the oxidation potential of DA was smaller than that of AA. Our simulations also showed that UA had the highest affinity for the GO surface binding and the smallest number of hydrogen bonds formed with serine oligomers. The electron leaking from UA to the epoxy group of GO through pi-orbitals resulting from their high binding preference was also proposed at the DFT level as the mechanism against the oxidation reactions [35]. Therefore, the oxidation peak of UA in the experiment was the least affected, suggesting that the atomistic MD simulations agreed with the oxidation peak from a differential pulse voltammetry.

Agreement of the in vitro and in silico results demonstrated the importance of the intrinsic properties, e.g., local polarity of functional groups, on the binding specificity and sensitivity of the electrode to the analytes. In addition, the simulation techniques had been validated for other designs of selective layers to detect other analytes, based-on the electro-polymerization of other biopolymers. However, not all molecular features of the real electrode system in vitro were depicted in the current in silico molecular model due to the limited simulation computing resource. Calibration of the model with the larger experimental dataset was also needed for more accurate prediction. 

## 4. Computational Methods

The 3D structure of a model graphene oxide nanosheet with dimensions of 20.94 nm × 22.56 nm in Figure 1a consisted of 160 carbon atoms, 72 hydrogen atoms, and 80 oxygen atoms were create by using the Avogadro program developed by Hanwell et al. [43]. The epoxy and hydroxyl groups were attached to the graphene plane according to the most energetically favorable configurations proposed by Yan and Chou [44]. The structure files of AA, DA, and UA analytes (Figure 1b–d) were then obtained from the PubChem database supported by the National Center for Biotechnology Information, US [45]. AA and UA were deprotonated into anionic forms at pH 7. Then, the topology files based on the Gromos54a7 force field were created for graphene oxide and analyte molecules by the Automated Topology Builder webservice supported by the National Computational Infrastructure (NCI), Australia [46]. A Tetraserine model was also created by Avogadro, and the forcefield file was created by the pdb2gmx suite embedded in the GROMACS 5.1.2 software package by research teams in the University of Groningen and Uppsala University [47]. Six starting structures for atomistic MD simulations were set: (a) GO + AA, (b) GO + DA, (c) GO + UA, (d) GO + SE + AA, (e) GO + SE + DA, and (f) GO + SE + UA. The first three structures consisted of a graphene oxide sheet and twelve analyte molecules within the 6 × 6 × 6 nm^3^ SPC [48] solvated simulation boxes. For the last three structures of the tetraserine decorated systems, a short simulation was performed to create a configuration that tetraserine molecules were adsorbed onto the graphene oxide prior to the addition of analytes. After energy minimization and pre-equilibration simulations for 100 ps in the NVT ensemble at 300 K using a Berendsen thermostat [49], each system underwent a 100-ns productive run in an NPT ensemble, which used a Berendsen thermostat and an isotropic Parrinello–Rahman barostat [50] at 300 K and 1 bar, respectively. The timestep was set to 2 fs and the trajectories were recorded every 10 ps. The cutoff radius for the vdW interaction was set to 1.0 nm. All the simulations were performed with GROMACS 5.1.2 [47]. 

After all simulations were completed, water molecules were deleted from the trajectory files. Minimum distances were measured between the closest atom pairs from each analyte molecule and the GO surface to monitor the adsorption of each molecule on the bare GO and the tetraserine-decorated GO surface. Then, radial distribution functions (RDF) and coordination numbers (Cn) were calculated as a function of closest distance from the GO surface for the GO + AA, GO + DA, and GO + UA systems without serine tetramers. RDF and Cn of analyte molecules were calculated both from the GO surface and from serine tetramers for the GO + SE + AA, GO + SE + DA, and GO + SE + UA systems decorated by serine tetramers. To further quantify the adsorption of analyte molecules, the binding constant (k) was calculated from the cumulative average number of analytes at the first adsorption layer (N) by using Equation (1). Numbers of hydrogen bonds formed (1) between analytes and GO and (2) between tetraserine and GO were also calculated along all the trajectories. The cut-off distance between the donor atom and the acceptor atom of a hydrogen bond was set to 0.35 nm, and the bending angle cut-off was set to 30°. Finally, the RDF between hydrogen bond donor atoms of the analytes and all the oxygen atoms of the tetraserines was also calculated to display the contribution of each atom to selective binding of the analytes on the tetraserine surface.

## Figures and Tables

**Figure 1 molecules-26-02876-f001:**
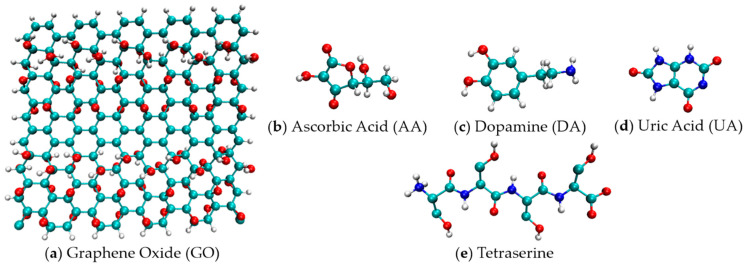
Atomistic structures of the molecular models used in this study: (**a**) graphene oxide; (**b**) ascorbic acid; (**c**) dopamine; (**d**) uric acid; and (**e**) tetraserine.

**Figure 2 molecules-26-02876-f002:**
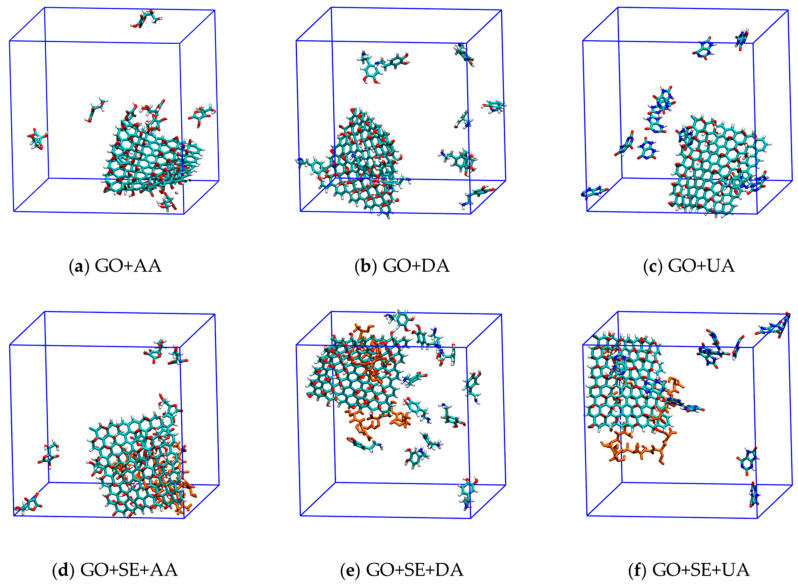
Starting structures for all atomistic MD simulations: (**a**) GO + AA; (**b**) GO + DA; (**c**) GO + UA; (**d**) GO + SE + AA; (**e**) GO + SE + DA; and (**f**) GO + SE + UA.

**Figure 3 molecules-26-02876-f003:**
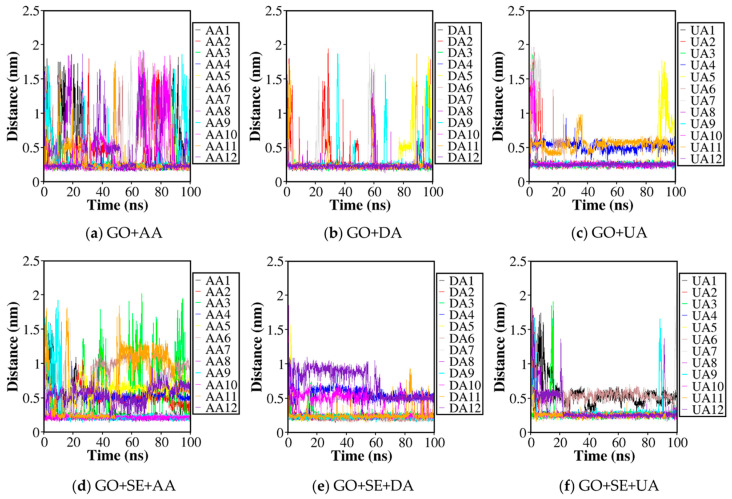
Minimum distances measured between each of the twelve analyte molecules and GO in the (**a**) GO + AA; (**b**) GO + DA; (**c**) GO + UA; (**d**) GO + SE + AA; (**e**) GO + SE + DA; and (**f**) GO + SE + UA systems. The final structure is also provided for each system. GO nanosheets are represented in grey and tetraserines are represented in orange.

**Figure 4 molecules-26-02876-f004:**
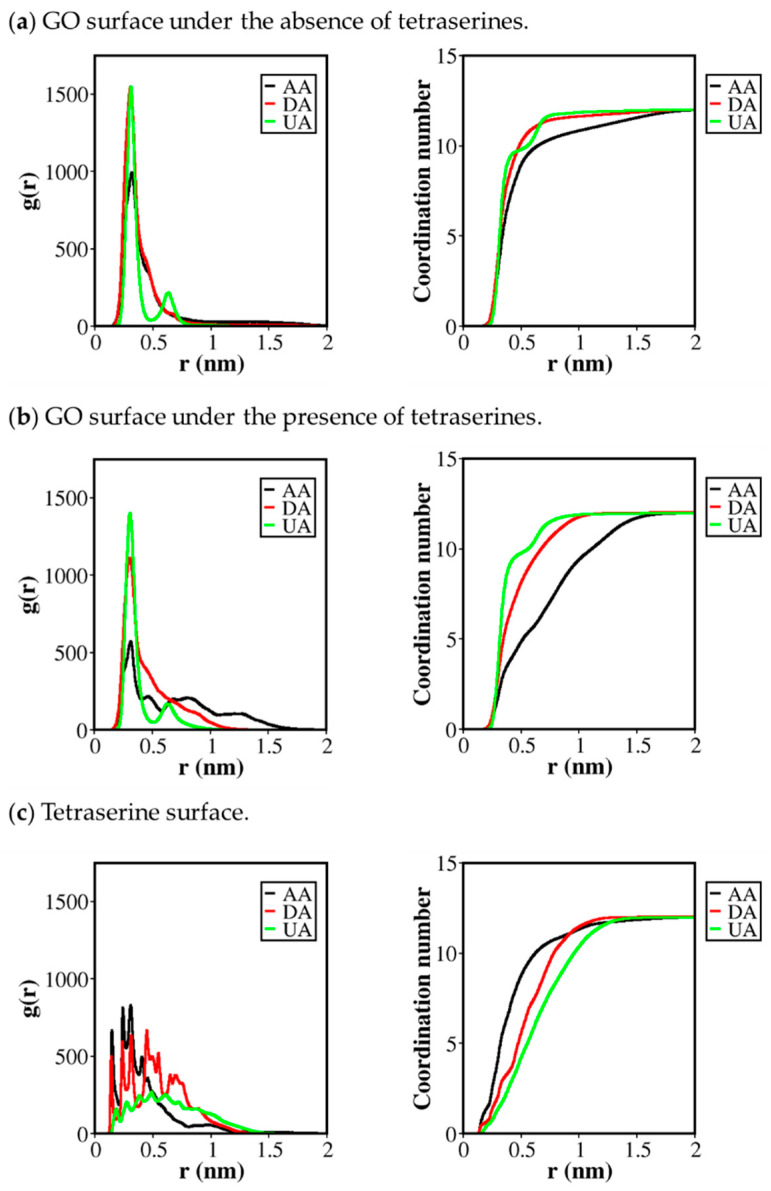
Radial distribution functions of AA/DA/UA analytes (**a**) about the GO surface under the absence of tetraserines; (**b**) about the GO surface under the presence of tetraserines; (**c**) about the tetraserines. Corresponding coordination numbers were also plotted.

**Figure 5 molecules-26-02876-f005:**
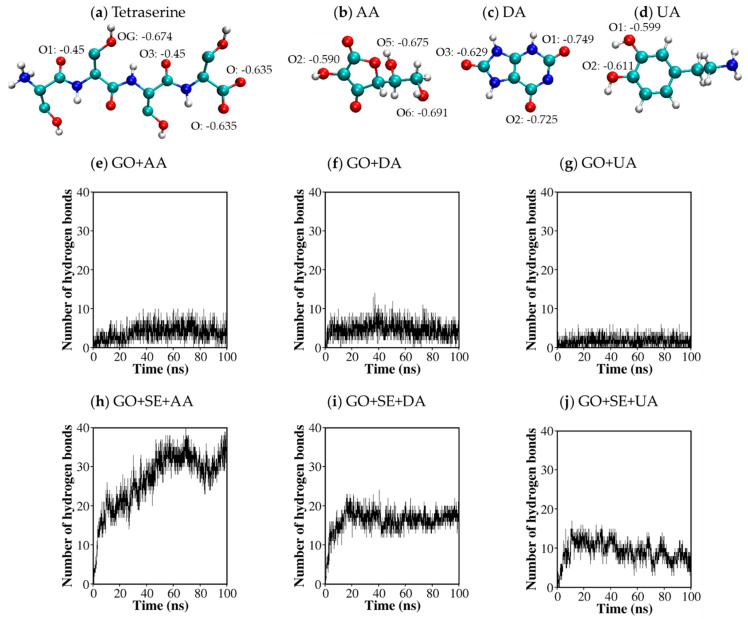
(**a**) Atomistic structures and partial charges within a tetraserine molecules in accordance with the GROMOS54a7 forcefield; (**b**–**d**) atomistic structures and partial charges of (**b**) AA; (**c**) DA; and (**d**) UA molecules in accordance with the GROMOS54a7 forcefield and 631G* DFT calculations; (**e**–**g**) number of hydrogen bonds between the (**e**) AA; (**f**) DA; and (**g**) UA analytes and GO; (**h**,**i**) number of hydrogen bonds between the (**h**) AA; (**i**) DA; and (**j**) UA analytes and tetraserines.

**Figure 6 molecules-26-02876-f006:**
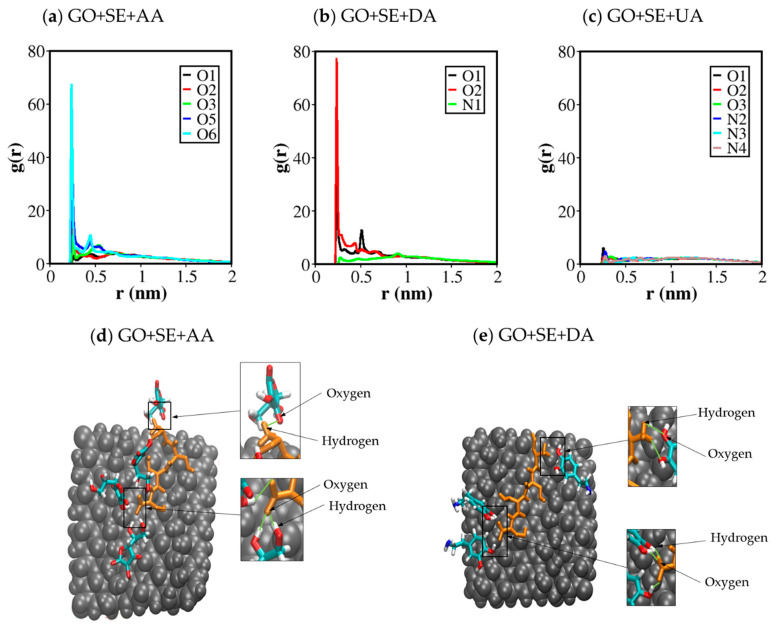
Radial distribution function (RDF) of each hydrogen bond donor atom with (**a**) AA; (**b**) DA; and (**c**) UA around the group of acceptor atoms within tetraserines; (**d**,**e**) a conformational snapshot taken after 100 ns showing hydrogen bonds formed by (**d**) AA and (**e**) DA on the tetraserine.

**Table 1 molecules-26-02876-t001:** Binding constant for GO-AA, GO-DA, GO-UA, GO-SE-AA, GO-SE-DA, and GO-SE-UA system.

System	Binding Constant on GO(M^−1^)	Binding Constant on Tetraserine(M^−1^)
GO-AA	0.62	-
GO-DA	1.16	-
GO-UA	1.03	-
GO-SE-AA	0.15	0.58
GO-SE-DA	0.45	0.18
GO-SE-UA	1.01	0.11

## Data Availability

Not applicable.

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
