# Peer review of "Molecular Mechanisms on the Selectivity Enhancement of Ascorbic Acid, Dopamine, and Uric Acid by Serine Oligomers Decoration on Graphene Oxide: A Molecular Dynamics Study"

_molecules, 2021, doi:10.3390/molecules26102876_

Round 1

Reviewer 1 Report

Molecules-1216788

The authors have made a thorough experiment on enhancing selectivity and detection of target molecules such as ascorbic acid, dopamine and uric acid by employing serine-coated graphene oxidewebs in a computational molecular dynamic simulation. Format and language seem well. The work is generally well-written and sound. However, some corrections should be made in order to improve its quality.

Introduction

The section is clear, cohesive and well structured. Nevertheless, there is a misuse of references as in just in a span of 34 lines there are already 45 citations. While these references appear to fit the topic and originate from reliable sources, consider limiting and relocating citations, also avoiding citing more than 3 works in the same citation. Additionally, a brief, yet more extensive elaboration would be needed on: how and why these target molecules are relevant for detection, to what diseases they are related to, comparison of conventional/current detection methods with graphene based-methods and how the novelty of this technique can open new possibilities. Moreover, a brief introduction on why serine oligomers were chosen would be desirable.

Figure 1: Check format, remove bars and capitalize the first letter of the title and structures. Repeat for all figures.

Computational methods

In addition to referencing them, state software and database proprietors.

Results

Figure 2: Same corrections as for Figure 1.

It would be best to include described methods (L91-93, L151-166, L273-L276, etc) in the proper Methods section.

L138: Check format.

Figure 3: Same corrections as for figures 1 & 2. However, the figure’s current size hampers interpretation, due to the format of the graphs. Consider dividing it into several figures and/or making them fit the whole page.

Figure 4: Albeit easier to read, size remains an issue. Magnify it and make the same changes previously suggested for the rest of the figures.

Figure 5: Same corrections as suggested for the rest of the figures.

Figure 6: Same corrections suggested, and graphs would need a bigger size to be easily read.

Discussion

Although English is generally excellent throughout the whole manuscript, it is a bit lacking in this section (i.e. “provided” instead of providing, redundant use of “the”, incorrect verb tenses, etc). I must advise a thorough check.

Nonetheless, although I think findings are generally well-discussed, scientific language and rigor should be improved, along with a proper discussion on the implications of these findings. How could they be applied? How would these results reflect on in vitro or in vivo experiments? Moreover, although the comparison is established with a previous report using tyrosine, a more elaborated discussion with other methods would be needed. Finally, you must make it very clear that these findings are the result of in silico simulations and not experiments carried out with the selected materials. Hence, they should be taken with caution.

Conclusions

Same thoughts for this section, as it is almost non-informative. Not only an abbreviation is once again introduced but the whole section is not meaningful. A very brief summary of your work and findings is not sufficient for conclusions. Rewrite it entirely.

Author Response

Reviewer 1

Comments and Suggestions for Authors

                The authors have made a thorough experiment on enhancing selectivity and detection of target molecules such as ascorbic acid, dopamine, and uric acid by employing serine-coated graphene oxidewebs in a computational molecular dynamic simulation. Format and language seem well. The work is generally well-written and sound. However, some corrections should be made in order to improve its quality.

Response: All authors thank the reviewer for his/her time and comments.

Introduction: The section is clear, cohesive and well structured. Nevertheless, there is a misuse of references as in just in a span of 34 lines there are already 45 citations. While these references appear to fit the topic and originate from reliable sources, consider limiting and relocating citations, also avoiding citing more than 3 works in the same citation. Additionally, a brief, yet more extensive elaboration would be needed on: (i) how and why these target molecules are relevant for detection, to (ii) what diseases they are related to, (iii) comparison of conventional/current detection methods with graphene based-methods and (iv) how the novelty of this technique can open new possibilities. Moreover, a brief introduction on why (v) serine oligomers were chosen would be desirable.

Response: Regarding the style, we tried to relocate the citations as suggested. Regarding the content of the intro, we here respond as following: (i) AA/DA/UA are relevant for detection with graphene-base electrodes due to their ring structure and their capability of being oxidized (line 38-40). (ii) We have added the details of the relevant diseases (see line 41-45). (iii) We have now added a few sentence comparing electrochem and other techniques by their physics principles (see line 51-57). (iv) We mentioned about the possibility of developing portable sensing platforms based-on the electrochem techniques (line 55-57). Finally, (v) we added that serine was with the highest density of uncharged polar groups, thus interact with polar functional groups of the analyte molecules through hydrogen bonds and facilitate redox reactions. (lines 71-74).    

Figure 1: Check format, remove bars and capitalize the first letter of the title and structures. Repeat for all figures.

Response: The changes on the figure have been made in the revised version.

Computational methods: In addition to referencing them, state software and database proprietors.

Response: We have now added more detail about two softwares and two databases/webservices mentioned in the Methods.

Figure 2: Same corrections as for Figure 1.

Response: The changes on the figure have been made in the revised version.

Results: It would be best to include described methods (L91-93, L151-166, L273-L276, etc) in the proper Methods section.

Response: We have now added more detail in the Methods section. (line 406-407, 410-415, 417-421)

Result: L138: Check format.

Response: This has been corrected. Thank you (line 155)

Figure 3: Same corrections as for figures 1 & 2. However, the figure’s current size hampers interpretation, due to the format of the graphs. Consider dividing it into several figures and/or making them fit the whole page.

Response: The changes on the figure have been made in the revised version.

Figure 4: Albeit easier to read, size remains an issue. Magnify it and make the same changes previously suggested for the rest of the figures.

Response: The changes on the figure have been made in the revised version.

Figure 5: Same corrections as suggested for the rest of the figures.

Response: The changes on the figure have been made in the revised version.

Figure 6: Same corrections suggested, and graphs would need a bigger size to be easily read.

Response: The changes on the figure have been made in the revised version.

Discussion: Although English is generally excellent throughout the whole manuscript, it is a bit lacking in this section (i.e. “provided” instead of providing, redundant use of “the”, incorrect verb tenses, etc). I must advise a thorough check.

Response: More grammatical corrections have been made in the revised version.

Discussion: Nonetheless, although I think findings are generally well-discussed, scientific language and rigor should be improved, along with a proper discussion on the implications of these findings. (i) How could they be applied? (ii) How would these results reflect on in vitro or in vivo experiments? Moreover, (iii) although the comparison is established with a previous report using tyrosine, a more elaborated discussion with other methods would be needed. Finally, (iv) you must make it very clear that these findings are the result of in silico simulations and not experiments carried out with the selected materials. Hence, they should be taken with caution.

Response: (i) the application of this finding is the use of computational tools to design the selective layers of other electrodes (line 372-374) (ii) we have now mentioned the relationship between the negative shifting of oxidation peak and the number of hydrogen bonds (line 326-329). (iii) we also mentioned a previous study (line 365-367). (iv) This has been stated at the end of the discussion paragraph (line 368-369).

Conclusions: Same thoughts for this section, as it is almost non-informative. Not only an abbreviation is once again introduced but the whole section is not meaningful. A very brief summary of your work and findings is not sufficient for conclusions. Rewrite it entirely.

Response: We have entirely deleted this section as it might be redundant. The information about the potential applications of the modeling ideas was in the latter part of the Discussion section. (line 370-377)

Reviewer 2 Report

Molecular Mechanisms on the Selectivity Enhancement of

Ascorbic Acid, Dopamine, and Uric Acid by Serine Oligomers

Decoration on Graphene Oxide : A Molecular Dynamics Study

This work of molecular dynamics (MD) simulations were performed to investigate the role of serine oligomers on the selectivity of the AA, DA, UA analytes. Our models consisted of a graphene oxide (GO) sheet under a solvent environment. Serine tetramers were added into the simulation box and were adsorbed on the GO surface.

Abstrac:  

You need to detail the results, it is necessary to be a little more specific.

Introduction:

Are there previous studies of separate calculations with graphene in the literature? Calculations with other levels of theory have been reported, express them in this section.

Prasert, K., & Sutthibutpong, T. (2021). Unveiling the Fundamental Mechanisms of Graphene Oxide Selectivity on the Ascorbic Acid, Dopamine, and Uric Acid by Density Functional Theory Calculations and Charge Population Analysis. Sensors, 21(8), 2773.

Tan, C., Zhao, J., Sun, P., Zheng, W., & Cui, G. (2020). Gold nanoparticle decorated polypyrrole/graphene oxide nanosheets as a modified electrode for simultaneous determination of ascorbic acid, dopamine and uric acid. New Journal of Chemistry, 44(12), 4916-4926.

Results and discussion

Not comment

Materials and Methods

Not comment

Conclusion

The conclusions are weak, it is suggested to reformulate, highlighting the work done and the data obtained.

Author Response

Comments and Suggestions for Authors

Molecular Mechanisms on the Selectivity Enhancement of Ascorbic Acid, Dopamine, and Uric Acid by Serine Oligomers Decoration on Graphene Oxide : A Molecular Dynamics Study

This work of molecular dynamics (MD) simulations were performed to investigate the role of serine oligomers on the selectivity of the AA, DA, UA analytes. Our models consisted of a graphene oxide (GO) sheet under a solvent environment. Serine tetramers were added into the simulation box and were adsorbed on the GO surface.

Response: All authors thank the reviewer for his/her time and comments.

Abstract:  You need to detail the results, it is necessary to be a little more specific.

Response: We have added some sentences in the Abstract as highlighted in red, citing more detail of the results and the discussion on the mechanisms.

Introduction: Are there previous studies of separate calculations with graphene in the literature? Calculations with other levels of theory have been reported, express them in this section.

Response: The work is now mentioned in line 82-84 and was discussed again in the discussion section (line 365-367). The work cited was so far the only molecular modeling work on the graphene interactions with AA/DA/UA we can find.

Conclusion: The conclusions are weak, it is suggested to reformulate, highlighting the work done and the data obtained.

Response: We have entirely deleted this section as it might be redundant. The information about the potential applications of the modeling ideas was in the latter part of the Discussion section. (Line 370-377)
